# Rapid and Simultaneous Determination of Anabolic Andro-Genic Steroids in Livestock and Poultry Meat Using One-Step Solid-Phase Extraction Coupled with UHPLC–MS/MS

**DOI:** 10.3390/molecules29010084

**Published:** 2023-12-22

**Authors:** Liqun Wang, Yonghong Yan, Yan Wang, Qingqin Lv, Shuang Teng, Wei Wang

**Affiliations:** 1College of Food Science and Technology, Nanjing Agricultural University, Nanjing 210095, China; 2022108067@stu.njau.edu.cn (L.W.); 2021108044@stu.njau.edu.cn (Y.Y.); F2014021@njau.edu.cn (Q.L.); tengshuang@njau.edu.cn (S.T.); 2State Key Laboratory of Meat Quality Control and Cultured Meat Development, Nanjing Agricultural University, Nanjing 210095, China; 3Center of Agro-Product Safety and Quality, Ministry of Agriculture and Rural Affairs, Beijing 100125, China; wy5082@126.com

**Keywords:** anabolic androgenic steroids, one-step solid-phase extraction, UHPLC–MS/MS, livestock and poultry meat

## Abstract

Anabolic androgenic steroids (AASs) are usually illegally added to animal feed because they can significantly promote animal growth and increase carcasses’ leanness, which threatens the safety of animal-derived foods and indirectly hazards human health. This study aimed to establish an ultra-high performance liquid chromatography-tandem mass spectrometry (UHPLC–MS/MS) method for the simultaneous detection of twelve AAS residues in livestock and poultry meat. The homogenized samples were extracted with acetonitrile containing 1% acetic acid (*v*/*v*) and purified using the one-step extraction column. After concentration using nitrogen, the residues were redissolved in acetonitrile and then quantified with an external standard method using UHPLC–MS/MS. The results showed that the above-mentioned method had a satisfactory linear correlation (*R*^2^ ≥ 0.9903) with a concentration range of 1–100 μg/L, and the limits of detection (LODs) and quantification (LOQs) were 0.03–0.33 μg/kg and 0.09–0.90 μg/kg, respectively. With the intraday and interday precision less than 15%, the average recoveries of pork, beef, lamb, and chicken, at different spiked levels, ranged from 68.3 to 93.3%, 68.0 to 99.4%, 71.6 to 109.8%, and 70.5 to 97.7%, respectively. Overall, the established method is validated, precise, and capable of the high-throughput determination of the residues of twelve AASs in livestock and poultry meat.

## 1. Introduction

Anabolic androgenic steroids (AASs) are a class of synthetic compounds with cyclopentane polyhydrophenanthrene nuclei that are strongly bound to globulin and weakly bound to albumin. They can participate in anabolism in humans, with the effects of increasing muscle strength and decreasing fat content [1,2,3]. Steroid hormones are illegally added to feed by farmers to promote animal growth and increase carcasses’ leanness, with AASs being the more commonly used class of prohibited drugs [4,5,6]. AASs are difficult to degrade and easily accumulate in animals. If humans consume meat products containing the residues of AASs for a long period, a series of symptoms, such as cardiovascular diseases, endocrine disorders, and cancer, can be induced [7,8,9,10,11]. The abuse of AASs has become a potential health problem in animal-derived foods. Many organizations and countries have made relevant regulations on the residues of AASs in animal-derived foods, such as the European Union (EU), China, Canada, the United States, and Japan, which prohibit the use of AASs in animal feed [12]. As an important part of the human diet, the quality and safety of meat is related to the health of consumers and the import and export trade. Therefore, it is particularly important to establish sensitive, efficient, and high-throughput methods for the detection of AAS residues in livestock and poultry meat.

Livestock and poultry meat contain endogenous substances, such as proteins and lipids, that can interfere with AAS residue detection [13]. Therefore, the selection of a suitable purification method is one effective means of ensuring the accuracy of quantification. Current methods for purification and enrichment of AASs include liquid-liquid extraction (LLE) [14], the Quick, Easy, Cheap, Effective, Rugged, and Safe (QuEChERS) method [15], magnetic solid-phase extraction (MSPE) [16], and solid-phase extraction (SPE) [17], etc. Among them, the LLE method consumes a lot of organic reagents which are harmful to the environment. The QuEChERS method requires the lesser usage of organic reagents, but it is unsuitable for the purification of fatty samples. Moreover, the MSPE technique causes the low recovery of meat products. By contrast, the SPE method has the advantages of high stability, high recovery, high maneuverability, and applicability to large-volume sample processing, etc. Coupled with the increasing variety of SPE columns, SPE is the most commonly used method for sample purification today. Therefore, in this study, the SPE method was selected to purify and enrich AASs in livestock and poultry meat.

Analytical methods can directly affect the sensitivity, throughput, and accuracy of the established method. Therefore, it is important to choose suitable analytical methods for the quantification of AAS extracts. Currently, the commonly used analytical methods for AASs in animal-derived foods include enzyme-linked immunosorbent assay (ELISA) [18], gas chromatography-tandem mass spectrometry (GC–MS) [19], liquid chromatography-tandem mass spectrometry (LC–MS/MS) [20], and ultra-high performance liquid chromatography-tandem mass spectrometry (UHPLC–MS/MS) [21], etc. ELISA with limited sensitivity is mostly used for routine screening. Instrumental methods, such as GC–MS, LC–MS/MS, and UHPLC–MS/MS, have high sensitivity [22]. Compared with the above detection methods, strong quasi-molecular ion peaks of the components to be measured can be obtained using UHPLC–MS/MS under primary mass spectrometry conditions. Furthermore, the quasi-molecular ions can be further cleaved to obtain the ion peaks, which can then exclude matrix interference and accurately identify the target compounds. It also has the significant advantages of high sensitivity, low injection volume, low matrix interference, and high throughput, and has become the mainstream for the trace analysis of steroid hormones in recent years [23,24]. However, based on the current UHPLC–MS/MS detection technology, complex pretreatment of samples is required, which consumes a lot of organic solvents and time. In this study, by optimizing the pretreatment and instrument conditions, an accurate, rapid, and high-throughput detection method of twelve AASs (the structural formula is shown in Figure 1) in livestock and poultry meat was established with UHPLC–MS/MS. The establishment of this method is not only of great significance for safeguarding meat safety, but also provides a reference for animal-derived food monitoring, routine batch testing, and confirmatory analysis of AASs in livestock and poultry meat.

## 2. Results and Discussion

### 2.1. Stability Testing of Standard Working Solutions

In order to ensure the accuracy of the experimental results, the standard working solution (5 mg/L) was subjected to a stability test for a period of thirty-five days. The test was performed once a week during storage, and the determination was conducted in triplicate. *t* tests were performed on the assay results using IBM SPSS Statistics, as shown in Figure 2, which indicated that all results were within the 95% confidence interval (average = 4.994, *p* = 0.685), suggesting that the standard working solution was stable during thirty-five days.

### 2.2. Optimization of UHPLC–MS/MS Conditions

#### 2.2.1. Optimization of Chromatographic Condition

Testosterone and epitestosterone, as well as metenolone and methyltestosterone, were isomers of each other, which increased the difficulty of chromatographic separation. According to previous studies, a reversed-phase column was recommended to separate AASs [25,26]. Therefore, the separation performances of three columns, namely: ACQUITY UPLC BEH C18 (1.7 μm, 2.1 mm × 100 mm), Thermo Hypersil GOLD aQ (1.9 μm, 100 mm × 2.1 mm), and Shim-pack GIST-HP C18-AQ (1.9 μm, 2.1 mm × 100 mm), were compared. According to the results, the Shim-pack GIST-HP C18-AQ (1.9 μm, 2.1 mm × 100 mm) was selected for its stable baseline, narrow front, highest separation efficiency, and best separation of the target compounds.

In order to better separate the twelve AASs, different mobile phases, such as 0.1% aqueous formic acid *(v*/*v)* containing 2 mM ammonium acetate–methanol, 0.1% aqueous formic acid *(v*/*v)* containing 2 mM ammonium acetate–acetonitrile, 0.1% aqueous formic acid *(v*/*v)*–acetonitrile containing 0.1% formic acid *(v*/*v)*, and 0.1% aqueous formic acid *(v*/*v)*–acetonitrile, were tried. The results showed that the addition of formic acid and ammonium acetate enhanced the signal responsiveness of the target compounds, which is in line with the reports [27,28]. Moreover, acetonitrile is less viscous than methanol, and is the same solvent as the dilution of the standard, so there is less detection noise, low solvent effect, and a sharp symmetry of the fronts. In summary, 0.1% aqueous formic acid (*v*/*v*) containing 2 mM ammonium acetate–acetonitrile was selected as the mobile phase.

Using isocratic elution was difficult to achieve chromatographic separation due to the similar polarity of the twelve compounds. Therefore, the gradient elution procedure was determined by optimizing the elution intensity of mobile phases, and the analysis time was 5 min, which is faster than the method in the Chinese National Standard GB/T 21981-2008 for the detection of multiple residues of hormones in 18 min [29]. The developed method can separate the co-eluting peaks (say critical pairs 2 and 3, 5 and 6, 9 and 10) using MRM mode in mass spectrometry, and the ultra-high performance liquid chromatogram of the mixed solution is shown in Figure 3.

#### 2.2.2. Optimization of Mass Spectrometry Condition

Twelve single standard solutions of 100 μg/L were injected into the mass spectrometer at a flow rate of 5 μL/min by using a needle pump. The Q1 full scan was turned on in positive ion mode to determine the molecular ion of each standard. Then, the instrument was scanned in SIM Q3; for each target compound, the two pairs of characteristic ion pairs with the highest response were selected as qualitative and quantitative ion pairs (Appendix A). Various mass spectral parameters were optimized in the MRM mode (Table 1).

### 2.3. Optimization of Pretreatment Conditions

#### 2.3.1. Optimization of Extraction Solvent

AASs belong to the medium-polar compounds, which are easily soluble in moderately polar organic solvents. By reviewing the literature, the commonly used extraction solvents are acetonitrile [30], ethyl acetate [31,32], methanol [33], and acetonitrile containing 1% acetic acid (*v*/*v*) [28]. Based on the spiked recovery of twelve AASs, the effects of methanol extraction, acetonitrile extraction, ethyl acetate extraction, and acetonitrile containing 1% acetic acid *(v*/*v)* extraction were compared. As shown in Table 2, the recoveries of acetonitrile, ethyl acetate, and methanol as extraction solvents did not meet the requirements in the range of 60–120%. Additionally, the extraction system of acetonitrile containing 1% acetic acid (*v*/*v*) was less interfering in the assay compared to methanol and ethyl acetate. Acetonitrile containing 1% acetic acid gave the best extraction, because the addition of a tiny quantity of acid to acetonitrile favored protein precipitation, which is consistent with the results reported in the literature [34,35]. In summary, samples without enzymatic treatment were extracted using acetonitrile containing 1% acetic acid, which improved the detection efficiency and met the requirements of the recovery.

#### 2.3.2. Optimization of Solid-Phase Extraction Column

There were still many impurities in the extracts that interfered with the instrumental response of the target compounds. The extracts needed to be further purified, to remove the endogenous substances and enrich the target compounds. The Oasis HLB column [30] (200 mg/3 mL, Waters, Milford, MA, USA), Bond ElutC18 column [36] (200 mg/3 mL, Agilent Technologies, Santa Clara, CA, USA), and Oasis MAX column [37] (60 mg/3 mL, Waters, Milford, MA, USA) were more commonly used in the assays of AASs. The QVet-NM column (5 mL–5 g, Shimadzu, Kyoto, Japan), as a new one-step column, is less utilized at present. Based on the spiked recoveries of twelve AASs, the purification effects of the above four SPE columns were compared. The results showed that the HLB, C18, and MAX SPE columns had low recoveries, and they required a series of operations, such as activation, equilibration, elution, and other steps that consumed a large amount of organic solvent. The QVet-NM column utilized pyrrolidinyl-modified polystyrene–divinylbenzene polymer packing for the targeted adsorption of phospholipid compounds from animal-derived samples with the best purification effect, and the recoveries of the target compounds all met the requirements. In summary, the QVet-NM SPE column was selected for solid-phase extraction in this study, and the effect of SPE columns on the recovery of the twelve AASs is shown in Table 3.

### 2.4. Validation of Bioanalytical Methods

#### 2.4.1. Matrix Effect Evaluation and Elimination

Matrix effects were evaluated at concentration levels from 1 to 100 μg/L. As shown in Figure 4, the calculated MEs are all less than 0.8. The results indicate that the twelve target compounds had matrix inhibition effects in four matrices, which interfered strongly with the detection of the target compounds. The matrix effects of the same compounds were different in different matrices. To ensure the accuracy of the established method, the matrix-matched standard curve was chosen to correct the matrix effect and improve the accuracy of the quantitative analysis results.

#### 2.4.2. Linearity of the Standards Curves, Limit of Detection, and Limit of Quantification

The prepared blank matrix-matched standard solutions for pork, beef, lamb, and chicken were detected separately based on the constructed UHPLC–MS/MS method. The results are shown in Table 4, and the matrix-matched curves of the twelve AASs showed good linearity in the concentration range of 1–100 μg/L with a square correlation coefficient (R^2^) ≥ 0.9903. According to the signal-to-noise (S/N), the limits of detection (LODs) and limits of quantification (LOQs) of the method ranged from 0.03 to 0.33 μg/kg, and the LOQs ranged from 0.09 to 0.90 μg/kg, which was lower than the LOQs of AASs in the Chinese National Standards [29,38]. The results indicated that the detection method had wide linearity, a low LOD, and good precision.

#### 2.4.3. Recovery and Precision

The mean recovery ranges for pork, beef, lamb, and chicken at different spiked levels were 68.3–93.3%, 68.0–99.4%, 71.6–109.8%, and 70.5–97.7%, respectively. The intraday and interday precision (RSDs) of all the compounds in the four matrices was 3.7–14.0% and 4.4–12.5%, respectively. Table A1, Table A2, Table A3 and Table A4 showed that the mean recoveries and precision of the twelve AASs in four matrices satisfy the requirements of a recovery range of 60–120% and a precision of less than 21% [39]. These results indicate that the method, which detects twelve AASs in livestock and poultry meat, has good accuracy and stability.

### 2.5. Analyses of Commercial Samples

During the detection of thirty-two commercial samples, the results showed that both testosterone and epitestosterone were detected in one beef sample (sample number: B-1), and the drug concentrations were 0.26 ± 0.02 μg/kg and 0.15 ± 0.01 μg/kg, respectively. The test result was judged to be positive, with a detection rate of 3.12%. The extracted ion chromatograms of the positive sample are shown in Figure 5. The blank samples were used as quality control (QC) samples by spiking during the assay. The standards were added at 5.0 μg/kg to the QC samples, and the recoveries of the QC samples ranged from 72.3 to 88.2%. The identification of the positive sample, and the validation results of the QC sample, further proved the feasibility and accuracy of the method.

## 3. Materials and Methods

### 3.1. Chemicals and Reagents

Twelve AAS standards, including testosterone, epitestosterone, methyltestosterone, nandrolone, boldenone, metandienone, trenbolone, metenolone, methandriol, mesterolone, danazol, and stanozolol were obtained from Alta Scientific Co., Ltd. (Tianjin, China). The concentration of all standards was 100 mg/L and their purity was ≥99%. Acetonitrile (HPLC-grade), methanol (HPLC-grade), and ethyl acetate (HPLC-grade) were purchased from Merck (Darmstadt, Germany). Acetic acid (HPLC-grade) was purchased from ACS Enkei Chemical Co., Ltd. (Shanghai, China). Sodium sulfate anhydrous was purchased from Shanghai McLean Biochemical Technology Co., Ltd. (Shanghai, China). Formic acid (HPLC-grade) and ammonium acetate (HPLC-grade) were purchased from Sinopharm Group Chemical Reagent Co., Ltd. (Shanghai, China). Ultrapure water was supplied by the Sartorius Arium^®^ Pro system (Sartorius, Göttingen, Germany), and SHIMSEN QVet-NM SPE columns were purchased from Shimadzu (Shimadzu, Kyoto, Japan).

### 3.2. Sample Collection

The Supervision, Inspection, and Testing Center for Quality of Meat Products (Nanjing, China) provided the blank pork, beef, lamb, and chicken samples used in the validation process. A total of thirty-two commercially available livestock and poultry meat products (sample numbers: P-1, P-2, P-3, P-4, P-5, P-6, P-7, P-8, B-1, B-2, B-3, B-4, B-5, B-6, B-7, B-8, L-1, L-2, L-3, L-4, L-5, L-6, L-7, L-8, C-1, C-2, C-3, C-4, C-5, C-6, C-7, C-8), tested in actual samples, were obtained from three different provinces in China. The letters P, B, L, and C stand for pork, beef, lamb, and chicken, respectively, and the numbers 1–3 represent samples from Jiangsu Province, 4–6 from Heilongjiang Province, and 7–8 from Guangxi Province.

### 3.3. Preparation and Stability Testing of Standard Solutions

Twelve standard solutions (100 mg/L) can be stored at −20 °C for up to twelve months, according to Alta Scientific Co. instructions. Standard working solutions were prepared by diluting twelve standard solutions with acetonitrile to achieve a concentration of 5 mg/L. Subsequently, these solutions were stored at −20 °C and subjected to stability testing over a period of thirty-five days (once a week). The mixed standard solutions, with a concentration of 100 µg/L, were obtained by diluting the aforementioned solutions with acetonitrile, and they were freshly prepared at the time of use.

### 3.4. UHPLC–MS/MS Instrumentation and Operating Conditions

Thermo Scientific Vanquish ultra-high performance liquid chromatography, coupled with the Thermo Scientific TSQ Quantis mass spectrometer (Thermo Fisher Scientific, Waltham, MA, USA), was used to identify the AASs. The Shim-pack GIST-HP C18-AQ column (1.9 μm, 2.1 mm × 100 mm, Shimadzu Corporation, Kyoto, Japan) equipped with the Shim-pack GIST-HP (G) C18-AQ column (1.9 μm, 2.1 mm × 10 mm, Shimadzu Corporation, Kyoto, Japan) was used to separate the AASs. The column temperature was 40 °C, and the sample injection volume was 10 μL. Mobile phase A was 0.1% aqueous formic acid *(v*/*v)* containing 2 mM ammonium acetate; mobile phase B was acetonitrile, and the flow rate was held at 0.3 mL/min. Gradient conditions were 0–0.5 min (40% B), 1.5–2.5 min (95% B), and 2.6–5 min (40% B). The total analysis time was 5 min.

The MS/MS was used in the multiple reaction monitoring (MRM) mode, and it was equipped with an ESI source operating in the positive ionization mode [M + H]^+^. The optimized electrospray ionization parameters were as follows: capillary voltage, 3.6 kV; ion source temperature, 450 °C; ion transfer tube temperature, 352 °C; vaporizer temperature, 350 °C; sheath gas, 30 Arb; auxiliary gas, 15 Arb.

### 3.5. Sample Preparation

Samples of pork, beef, lamb, and chicken (5.00 ± 0.02 g) were accurately weighed (SECURA313-ICN, Sartorius, Göttingen, Germany) into a 50 mL polypropylene centrifuge tube after being chopped and homogenized using an HM6300 intelligent homogenizer (Lab Precision Beijing Technology Co., Ltd., Beijing, China). The samples were extracted by adding 10 mL of acetonitrile containing 1% acetic acid *(v*/*v)*. Then, in order to remove water and promote the transfer of the targets to the acetonitrile phase, 5 g of anhydrous sodium sulfate was added [40,41,42]. Next, the sample systems were vortexed for 1 min and shaken for 10 min (HS 501 digital IKA^®^-WERKE, IKA, Staufen, Germany). Next, they were centrifuged at 10,621× *g* for 15 min at 4 °C (Centrisart^®^ D-16C, Sartorius, Göttingen, Germany), and the supernatant was transferred to another 50 mL polypropylene centrifuge tube. Then, 6 mL of supernatant was passed through the QVet-NM one-step solid-phase extraction column at a rate of about 1 drop/second for purification. Next, the filtrate was transferred to a 10 mL nitrogen blowing tube, and nitrogen blown to 50 μL for 25–30 min at 45 °C in a water bath (N-EV AP-11 nitrogen evaporator, Organization, Berlin, MA, USA). The residue was filled with acetonitrile to 0.5 mL. After vortex mixing for 1 min, the solution was collected in the injection bottle through a 0.22 μm hydrophilic filter membrane (Agilent Technologies, Santa Clara, CA, USA) for UHPLC–MS/MS analysis. The schematic diagram of the experimental procedure is shown in Figure 6.

### 3.6. Method Validation

The assessed parameters including matrix effects, linearity, sensitivity, precision, recovery, and analysis of commercial samples were evaluated in accordance with Appendix F of the Chinese National Standard GB 27404-2008 [38].

#### 3.6.1. Matrix Effect Evaluation

The substances in the meat matrix compete with the target compounds during ionization, leading to a decrease or increase in the signal responsiveness of the target compounds, so it is important to examine the matrix effects [20,27]. To analyze the matrix effects, the slopes of matrix-matched standard curves prepared from blank samples (the seven concentration levels of compounds were 1, 2, 5, 10, 20, 50, and 100 μg/L), and the reagent standard curves obtained by dilution with acetonitrile, were compared. The formula is:(1)$$\mathrm{ME}=\frac{k_{a}}{k_{b}}$$
where *k_a_* is the slope of the matrix-matched standard curve and *k_b_* is the slope of the reagent standard curve. When ME is between 0.8 and 1.2, the matrix effect is considered to be within the acceptable range; when ME < 0.8, it indicates a matrix inhibition effect; and when ME > 1.2, it indicates a matrix enhancement effect [34].

#### 3.6.2. Determination of Linearity, LODs, and LOQs

Based on the aforementioned UHPLC–MS/MS method, the matrix-matched standardized working solutions for pork, beef, lamb, and chicken were determined on the machine. The standard curve was plotted with the concentration (μg/L) of each target compound as the X-axis, and the peak area of the corresponding target compound as the Y-axis, to obtain the regression equation and calculate *R*^2^.

The LODs and LOQs for the twelve AASs were evaluated by testing a series of concentrations of spiked blank samples. The spiked concentration at S/N ≥ 3 was used as the LOD of the method, and the spiked concentration at S/N ≥ 10 was used as the LOQ of the method [43].

#### 3.6.3. Recovery and Precision Test

The established method’s accuracy (expressed as recovery) and precision (expressed as relative standard deviation, RSD) were tested with spiked blank samples at three concentration levels: 0.5, 1.0, and 5.0 μg/kg. For each concentration, six experiments were set up in parallel, and the analysis was repeated three times (on three different days). The average recovery was calculated along with intraday RSDs and interday RSDs. These results were used to evaluate the accuracy, precision, and stability of the established detection methods. The recovery and RSDs were calculated as follows:(2)$$\mathrm{recovery}\;\left(\%\right)=\frac{C_{E}}{4\times C_{S}}\times 100$$
(3)$$RSD\left(\%\right)=\frac{\mathrm{SD}}{C_{A}}\times 100$$
where *C_E_* (μg/L) is the experimental concentration determined from the calibration curve, 4 is the conversion multiplier, *C_S_* (μg/kg) is the spiked concentration, *SD* (μg/L) is the standard deviation, and *C_A_* (μg/L) is the average of the experimental concentration determined from the calibration curve.

### 3.7. Analysis of Commercial Samples

In order to verify the validity of the established method, a total of thirty-two samples of pork, beef, lamb, and chicken from different provinces of China were randomly selected and tested for twelve AAS residues, according to the method. Twelve compounds were quality-controlled by blank spiking to further validate the feasibility and accuracy of the method.

### 3.8. Data Analysis

TraceFinder 4.1 software (Thermo Fisher Scientific, Waltham, MA, USA) was used for data acquisition and processing; OriginPro 2023b software (version number: 10.0.5.157, OriginLab Inc., Northampton, MA, USA) was used for plotting; and IBM SPSS Statistics was used for significance analysis. Three parallels were performed for each experiment, and data were expressed as the mean ± standard deviation.

## 4. Conclusions

AASs have been illegally used in farming in recent decades due to their ability to promote animal growth and increase carcasses’ leanness. However, AASs can accumulate in animals and indirectly endanger human health. To meet the requirement for high-throughput detection of AAS residues in animal-derived foods, accurate and rapid testing methods should be developed and considered. In this study, a UHPLC–MS/MS method was established for the simultaneous detection of twelve AAS residues in livestock and poultry meat. Following extraction, the extracts were purified using the QVet-NM one-step column, which greatly improved the detection efficiency and consumed fewer organic solvents. Moreover, the pretreatment conditions were finished in all steps within four hours, and the UHPLC–MS/MS analysis time was only around 5 min, which improved the detection efficiency by about 65% compared with the current Chinese National Standards. Moreover, the results indicated that the twelve compounds showed good linearity in the concentration range of 1–100 μg/L with *R*^2^ ≥ 0.9903. The LODs of the method were in the range of 0.03–0.33 μg/kg, and the LOQs were in the range of 0.09–0.9 μg/kg. Both the intraday and interday precision were less than 15%, respectively. In conclusion, the established detection method in this paper has high sensitivity, reproducibility, and a short analytical time, and is capable of rapidly determining the trace residues of twelve AASs in livestock and poultry meat.

## Figures and Tables

**Figure 1 molecules-29-00084-f001:**
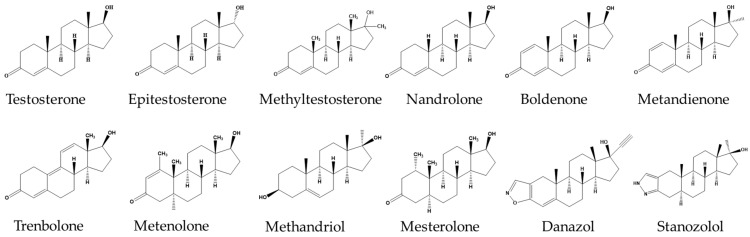
Structural formulas of twelve AASs.

**Figure 2 molecules-29-00084-f002:**
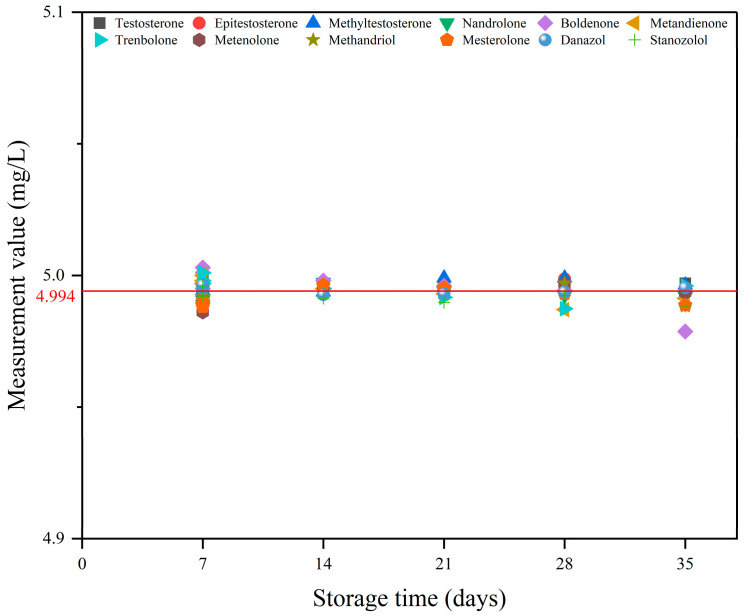
Scatter plot of stability analysis of standard working solutions.

**Figure 3 molecules-29-00084-f003:**
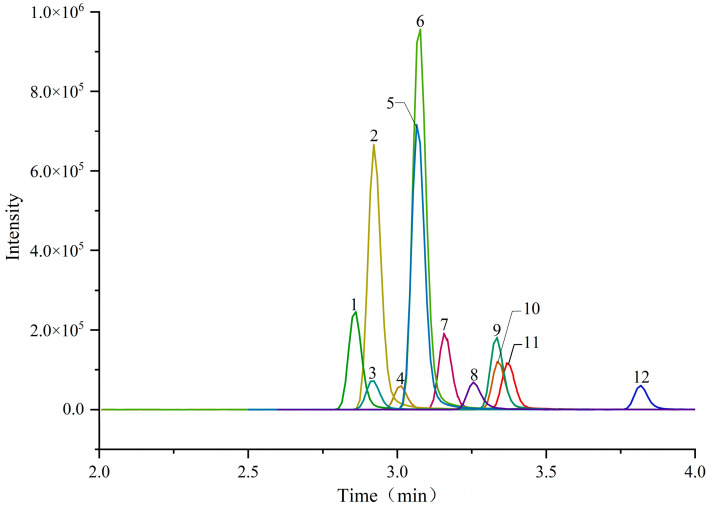
Ultra-high performance liquid chromatogram of the mixed solution (10 μg/L). Peaks: 1, Trenbolone; 2, Boldenone; 3, Methandriol; 4, Nandrolone; 5, Mesterolone; 6, Metandienone; 7, Testosterone; 8, Stanozolol; 9, Metenolone; 10, Methyltestosterone; 11, Epitestosterone; 12, Danazol.

**Figure 4 molecules-29-00084-f004:**
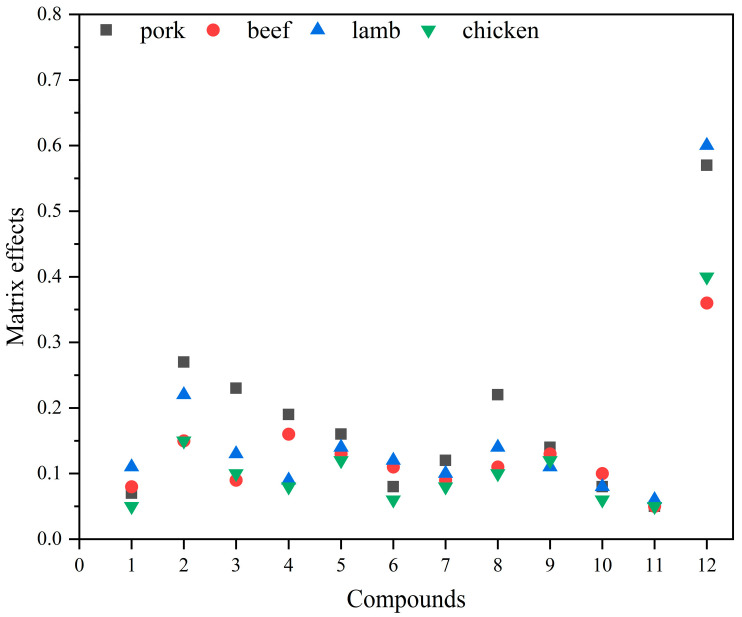
Matrix effects of twelve AASs in pork, beef, lamb, and chicken. X-axis 1–12 represent compounds as follows: 1, Testosterone; 2, Epitestosterone; 3, Methyltestosterone; 4, Nandrolone; 5, Boldenone; 6, Metandienone; 7, Trenbolone; 8, Metenolone; 9, Methandriol; 10, Mesterolone; 11, Danazol; 12, Stanozolol.

**Figure 5 molecules-29-00084-f005:**
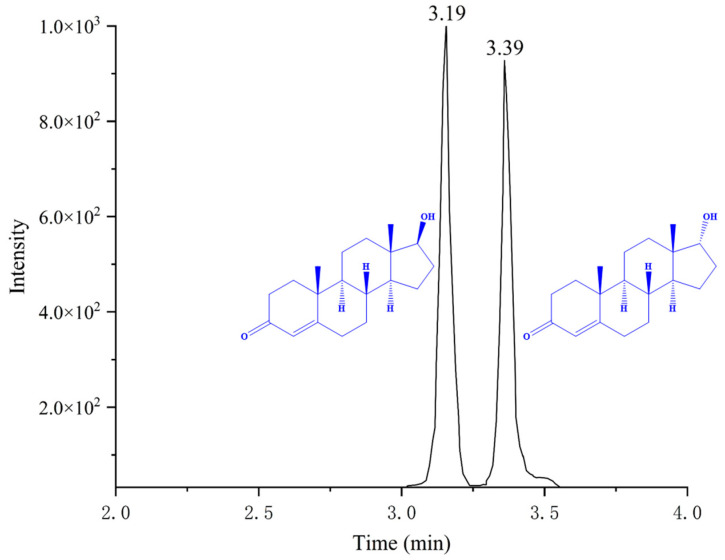
Extracted ion chromatogram of testosterone (3.19 min) and epitestosterone (3.39 min) in the positive sample.

**Figure 6 molecules-29-00084-f006:**
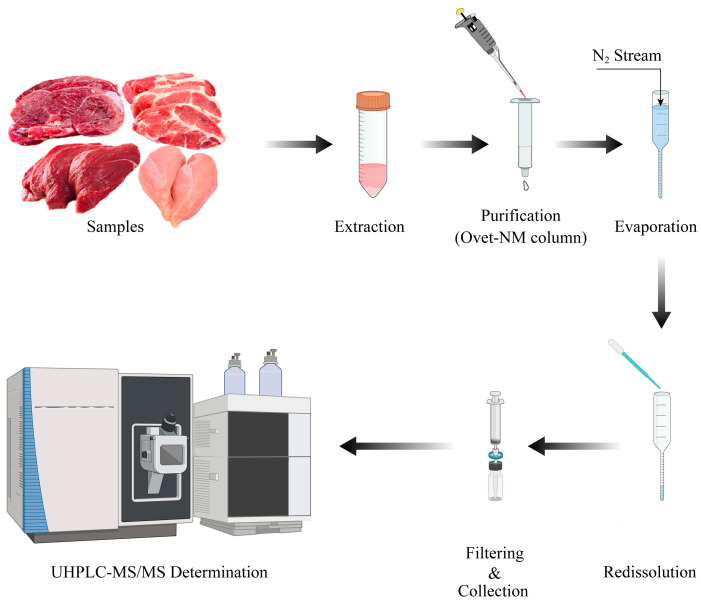
The schematic diagram of main experimental operation for the detection of AASs using UHPLC–MS/MS.

**Table 1 molecules-29-00084-t001:** Mass spectrometric analysis parameters of twelve AASs.

Compound	Precursor (*m*/*z*)	Product (*m*/*z*)	Collision Energy (V)	RF Lens (V)
Testosterone	289.212	97.000 *	21.68	120
	289.212	109.000	24.41	120
Epitestosterone	289.212	97.000 *	21.60	120
	289.212	109.000	24.37	120
Methyltestosterone	303.212	187.054 *	21.00	124
	303.212	285.137	16.00	124
Nandrolone	275.175	109.000 *	25.70	119
	275.175	257.208	15.50	119
Boldenone	287.175	121.071 *	22.85	95
	287.175	269.208	10.23	95
Metandienone	301.175	121.071 *	24.82	93
	301.175	149.155	14.32	93
Trenbolone	271.175	199.054 *	22.89	144
	271.175	253.137	19.52	144
Metenolone	303.212	109.000 *	26.14	124
	303.212	285.137	15.76	124
Methandriol	287.400	269.100 *	11.10	120
	287.400	159.100	21.10	120
Mesterolone	301.000	121.000 *	26.00	101
	301.000	149.000	15.00	101
Danazol	338.300	120.000 *	35.00	120
	338.300	148.200	15.00	120
Stanozolol	329.200	121.000 *	36.90	120
	329.200	107.100	42.00	120

*: quantitative ion.

**Table 2 molecules-29-00084-t002:** Effect of extraction solvent on the recovery of twelve AASs.

Extraction Solvent	Number of AASs		
	Recovery < 60%	Recovery 60% to 120%	Recovery > 120%
Methanol	5	6	1
Acetonitrile	1	8	3
Ethyl acetate	3	9	0
Acetonitrile (containing 1% acetic acid, *v*/*v*)	0	12	0

**Table 3 molecules-29-00084-t003:** Effect of SPE columns on the recovery of twelve AASs.

SPE Columns	Number of AASs		
	Recovery < 60%	Recovery 60% to 120%	Recovery > 120%
HLB	1	11	0
C18	2	10	0
MAX	4	8	0
QVet-NM	0	12	0

**Table 4 molecules-29-00084-t004:** Linear equations, LOD, and LOQ of twelve AASs in livestock and poultry meat.

Matrix	Compound	Regression Equation	R^2^	Linear Range (μg/L)	LOD (μg/kg)	LOQ (μg/kg)
Pork	Testosterone	*y* = 0.001690*x* − 0.001235	0.9957	1–100	0.08	0.24
	Epitestosterone	*y* = 0.002980*x −* 0.02382	0.9982	1–100	0.12	0.37
	Methyltestosterone	*y* = 0.002109*x* + 0.5309	0.9968	1–100	0.13	0.40
	Nandrolone	*y* = 0.09590*x* + 0.002411	0.9980	1–100	0.08	0.60
	Boldenone	*y* = 0.0001184*x −* 0.0001063	0.9966	1–100	0.06	0.18
	Metandienone	*y* = 0.007545*x* + 0.0001471	0.9995	1–100	0.03	0.09
	Trenbolone	*y* = 0.003224*x* + 0.006057	0.9984	1–100	0.12	0.36
	Metenolone	*y* = 0.002862*x* + 0.08988	0.9956	1–100	0.19	0.57
	Methandriol	*y* = 0.001485*x −* 0.3889	0.9968	1–100	0.24	0.72
	Mesterolone	*y* = 0.005878*x* + 0.006874	0.9929	1–100	0.03	0.09
	Danazol	*y* = 0.03880*x* + 0.07963	0.9977	1–100	0.30	0.90
	Stanozolol	*y* = 0.005689*x* + 0.0002292	0.9974	1–100	0.30	0.90
Beef	Testosterone	*y* = 0.001100*x* + 0.09015	0.9936	1–100	0.06	0.18
	Epitestosterone	*y* = 0.001678*x* + 0.003171	0.9979	1–100	0.06	0.18
	Methyltestosterone	*y* = 0.08854*x* + 0.004840	0.9962	1–100	0.07	0.23
	Nandrolone	*y* = 0.07633*x* + 0.004184	0.9986	1–100	0.09	0.30
	Boldenone	*y* = 0.009562*x* + 0.001751	0.9995	1–100	0.05	0.15
	Metandienone	*y* = 0.009745*x* + 0.01987	0.9915	1–100	0.03	0.09
	Trenbolone	*y* = 0.002319*x* + 0.09887	0.9985	1–100	0.09	0.30
	Metenolone	*y* = 0.001380*x* + 0.004296	0.9971	1–100	0.19	0.60
	Methandriol	*y* = 0.001413*x* + 0.07204	0.9968	1–100	0.27	0.80
	Mesterolone	*y* = 0.007392*x* + 0.004931	0.9934	1–100	0.03	0.09
	Danazol	*y* = 0.03512*x* + 0.09421	0.9985	1–100	0.17	0.50
	Stanozolol	*y* = 0.003588*x* + 0.0001100	0.9961	1–100	0.24	0.73
Lamb	Testosterone	*y* = 0.001527*x* + 0.006818	0.9995	1–100	0.05	0.15
	Epitestosterone	*y* = 0.002364*x* + 0.05624	0.9931	1–100	0.06	0.18
	Methyltestosterone	*y* = 0.001200*x* + 0.005994	0.9917	1–100	0.10	0.30
	Nandrolone	*y* = 0.04545*x* + 0.002813	0.9988	1–100	0.20	0.60
	Boldenone	*y* = 0.009871*x* + 0.006202	0.9957	1–100	0.05	0.15
	Metandienone	*y* = 0.0001146*x −* 0.006042	0.9992	1–100	0.29	0.90
	Trenbolone	*y* = 0.002577*x* + 0.002403	0.9987	1–100	0.03	0.09
	Metenolone	*y* = 0.001835*x* + 0.002530	0.9917	1–100	0.13	0.40
	Methandriol	*y* = 0.001440*x* + 0.006457	0.9977	1–100	0.26	0.80
	Mesterolone	*y* = 0.006078*x* + 0.001785	0.9979	1–100	0.03	0.09
	Danazol	*y* = 0.04350*x −* 0.08949	0.9903	1–100	0.33	0.90
	Stanozolol	*y* = 0.006099*x −* 0.002605	0.9972	1–100	0.09	0.30
Chicken	Testosterone	*y* = 0.07008*x* + 0.04804	0.9934	1–100	0.06	0.18
	Epitestosterone	*y* = 0.001597*x −* 0.002645	0.9970	1–100	0.08	0.24
	Methyltestosterone	*y* = 0.09237*x −* 0.02228	0.9964	1–100	0.13	0.40
	Nandrolone	*y* = 0.03718*x* + 0.001061	0.9912	1–100	0.19	0.60
	Boldenone	*y* = 0.008562*x −* 0.009533	0.9938	1–100	0.06	0.18
	Metandienone	*y* = 0.005442*x −* 0.0001624	0.9934	1–100	0.05	0.15
	Trenbolone	*y* = 0.002110*x −* 0.001296	0.9922	1–100	0.07	0.21
	Metenolone	*y* = 0.001328*x −* 0.09989	0.9965	1–100	0.09	0.30
	Methandriol	*y* = 0.001276*x* + 0.005650	0.9943	1–100	0.20	0.60
	Mesterolone	*y* = 0.003602*x −* 0.007355	0.9926	1–100	0.05	0.15
	Danazol	*y* = 0.03519*x −* 0.04157	0.9942	1–100	0.30	0.90
	Stanozolol	*y* = 0.003903*x −* 0.0001216	0.9929	1–100	0.20	0.60

## Data Availability

All available data are contained within the article.

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
