# Peer review of "Rapid and Simultaneous Determination of Anabolic Andro-Genic Steroids in Livestock and Poultry Meat Using One-Step Solid-Phase Extraction Coupled with UHPLC–MS/MS"

_molecules, 2023, doi:10.3390/molecules29010084_

Round 1
Reviewer 1 Report
Comments and Suggestions for Authors
The manuscript titled " Rapid and Simultaneous Determination of Anabolic Androgenic Steroids in Livestock and Poultry Meat Using One-Step 3 Solid-Phase Extraction Coupled with UHPLC-MS/MS" has been reviewed by me. There are still some issues that need to be addressed for it to be suitable for the MDPI publication. Hence, I recommend a major revision of this work.
Some specific comments can be found below.
1) While the abstract and introduction provide a substantial amount of background information related to anabolic androgen steroids, they lack the list of 12 analytes of interest.
2) Why external standard calibration was used instead of internal standard calibration, external standards do not correct for losses that may occur during the sample preparation, extraction, and sample evaporation?
3) What are the internal standards used for the current LC-MS bioanalysis study?
4) Mention the bioanalytical method validation guidelines that were used in the study
5) Sample stability is a major aspect in bioanalytical studies. Have the authors performed benchtop, long-term, and freeze-thaw stability testing? Please include this in the experimental section.
6) Since a biological matrix is involved kindly change the name to bioanalytical instead of analytical method, wherever applicable.
7) Accurate mass calibration of the instrument can be briefly summarized.
8) Page 3, line 104. Rewrite the sentence as the developed method can separate the co-eluting peaks (say critical pairs 2&3, 5&6, 9&10) using MRM mode in mass spectrometry.
9) Selectivity and carryover of the developed method must be proven by suitable evidence as carryover can give a false positive result.
10) Explain the role of sodium sulfate with references in sample preparation.
11) What is the rationale behind using 2mM ammonium acetate?
12) Provide the need for the conversion multiplier in Equation (2).
13) In the conclusion section, please add a sentence about QVet-NM extraction used for the purification of samples.
Minor comments:
14) Keep a single approved name of metenolone i.e., either methenolone or metenolone to avoid confusion for the readers and maintain consistency.
15) The full name of the term ‘Qvet-NM’ should be mentioned. Also, mention the source of procurement of SPE columns in Section 3.1.
16) In Section 2.4 and Figure 3, change the term ‘mass spectrogram’ to ‘extracted ion chromatogram’.
17) How was the uniformity in sample volume maintained under a gentle blow of nitrogen?
18) Slight dryness is a misleading terminology, mention the time frame for which samples were kept for evaporation to concentrate.
19) Please provide the precursor and product ion spectra (qualifier and quantifier) of all the androgenic steroids in a supplementary information.
Comments on the Quality of English Language
Can be improved
Author Response
Comment 1: While the abstract and introduction provide a substantial amount of background information related to anabolic androgen steroids, they lack the list of 12 analytes of interest.
Response 1: Thanks for your kind reminder. Structural diagrams of twelve AASs (Figure 1) has been added to the introductory section (Lines 83-84) and have been labeled in red.
Figure 1 Structural formulas of twelve AASs.
Comment 2: Why external standard calibration was used instead of internal standard calibration, external standards do not correct for losses that may occur during the sample preparation, extraction, and sample evaporation?
Response 2: Thank you very much for your comment and question. The analytical methodology elucidated in this study is proficient in both qualitative and quantitative analyses of samples through the utilization of the external standard method. This approach, deemed more suitable for the analysis and detection of a substantial volume of samples compared to the internal standard method, is characterized by its operational simplicity and computational convenience. Additionally, the reproducibility of the Thermo Scientific TSQ Quantis mass spectrometer has been ascertained. Monthly verification of the calibration curve is undertaken to ensure the steadfastness and repeatability of the external standard method. To address potential limitations of the external standard method, calibration efforts are directed at compensating for plausible losses incurred during sample preparation, extraction, and sample evaporation. This is achieved through the incorporation of matrix-matched standard working solutions.
Comment 3: What are the internal standards used for the current LC-MS bioanalysis study?
Response 3: Thanks for your question. To make it easier for you to review, I have summarized the internal standards used in Chinese National Standards for the detection of the AASs (Table 2) [1–4].
Table 2 Twelve AASs compounds and their internal standards.
|
Analyzing compounds |
Internal standards |
||
|
Compounds Name |
CAS |
Compounds Name |
CAS |
|
Testosterone |
58-22-0 |
Testosterone-2,2,4,6,6-d5 |
21002-80-2 |
|
Testosterone-16,16,17-d3 |
77546-39-5 |
||
|
Epitestosterone |
481-30-1 |
Testosterone-2,2,4,6,6-d5 |
21002-80-2 |
|
Methyltestosterone |
58-18-4 |
Testosterone-16,16,17-d3 |
77546-39-5 |
|
Nandrolone |
434-22-0 |
Testosterone-16,16,17-d3 |
77546-39-5 |
|
Boldenone |
846-48-0 |
Testosterone-16,16,17-d3 |
77546-39-5 |
|
d3-17β-Boldenone |
361432-76-0 |
||
|
Stanozolol |
10418-03-8 |
Testosterone-16,16,17-d3 |
77546-39-5 |
|
Metandienone |
72-63-9 |
Testosterone-16,16,17-d3 |
77546-39-5 |
|
Trenbolone |
10161-33-8 |
Testosterone-16,16,17-d3 |
77546-39-5 |
|
Metenolone |
153-00-4 |
Methyltestosterone-d3 |
96245-03-5 |
|
Danazol |
17230-88-5 |
Chlortestosterone-d3 acetate |
855-19-6 |
|
Mesterolone |
1424-00-6 |
Chlortestosterone-d3 |
69660-28-2 |
|
Methandriol |
521-10-8 |
Methyltestosterone-d3 |
96425-03-5 |
References:
[1]. Foodmate. Available online: http://down.foodmate.net/standard/yulan.php?itemid=17441 (accessed on 24 October 2023).
[2]. Foodmate. Available online: http://down.foodmate.net/standard/yulan.php?itemid=16931 (accessed on 24 October 2023).
[3]. Foodmate. Available online: http://down.foodmate.net/standard/yulan.php?itemid=11115 (accessed on 7 December 2023).
[4]. Foodmate. vailable online: http://down.foodmate.net/standard/yulan.php?itemid=126493 (accessed on 10 December 2023).
Comment 4: Mention the bioanalytical method validation guidelines that were used in the study.
Response 4: Thanks for your kind reminder. The test methods established in this paper are validated with reference to the test method validation techniques in Appendix F of Chinese National Standard GB 27404-2008[1]. And it is written in the experimental method section (Lines 293-295) and marked in red.
References:
[1]. Foodmate. Available online: http://down.foodmate.net/standard/yulan.php?itemid=15544 (accessed on 10 December 2023).
Comment 5: Sample stability is a major aspect in bioanalytical studies. Have the authors performed benchtop, long-term, and freeze-thaw stability testing? Please include this in the experimental section.
Response 5: Thanks for your kind reminder. We are very sorry for the misrepresentation of the use of the mixing label (100 ng/mL), which has been corrected and highlighted (Lines 252-253). In fact, the purchased standards (100 μg/mL) were divided into 250 μL and were ready-to-use to avoid the problem of freeze-thaw. The storage of the standards was carried out in strict accordance with the instructions, and the standards were used within their shelf life. What’s more, the stability of the standard working solutions at a concentration of 5 mg/L were tested for a period of thirty-five days during the test. The twelve standard working solutions were stable over thirty-five days (P > 0.05), and longer stability tests were not done because the standard dilutions could be used up within one month. Data were analyzed using IBM SPSS Statistics, P = 0.685, and compound concentrations were within the 95% confidence interval over thirty-five days’ stability test period. Stability experiments have been added to sections 2.1 (Lines 87-84) and 3.3 (Lines 249-251), and details have been labeled in red.
Comment 6: Since a biological matrix is involved kindly change the name to bioanalytical instead of analytical method, wherever applicable.
Response 6: Thanks for your kind reminder. The analytical methods in the manuscript have been changed to bioanalytical methods and labeled in red (Line 176).
Comment 7: Accurate mass calibration of the instrument can be briefly summarized.
Response 7: Thanks for your kind reminder. The instrument is calibrated and commissioned every 3-6 months and is calibrated annually in May by the Metrology Supervision and Inspection Institute of Nanjing, China. Figure 2 shows the calibration certificate (No. 01487423) for the Thermo Scientific TSQ Quantis, dated 2023.5.22.
Figure 2 The calibration certificate of the Thermo Scientific TSQ Quantis.
Comment 8: Page 3, line 104. Rewrite the sentence as the developed method can separate the co-eluting peaks (say critical pairs 2&3, 5&6, 9&10) using MRM mode in mass spectrometry.
Response 8: Thanks for your kind reminder. The original sentence has been rewritten as the developed method can separate the co-eluting peaks (say critical pairs 2&3, 5&6, 9&10) using MRM mode in mass spectrometry (Lines 122-124) and has been marked in red in the manuscript.
Comment 9: Selectivity and carryover of the developed method must be proven by suitable evidence as carryover can give a false positive result.
Response 9: Thanks for your kind reminder. The TSQ Quantis triple quadrupole mass spectrometer is highly selective and effective in avoiding false positives. Moreover, throughout the experimental proceedings, it is requisite to conduct a reagent blank sample analysis for every ten samples. In instances where the total number of samples is less than ten, a single reagent blank sample analysis shall be performed per batch.
Comment 10: Explain the role of sodium sulfate with references in sample preparation.
Response 10: Thanks for your kind reminder. In this study, we refer to the salting-out-acetonitrile homogeneous extraction method [1–3]. Sodium sulfate anhydrous removes water from the extract, promotes the separation of the aqueous and organic phases, facilitates the transfer of targets to the acetonitrile phase, and avoids water-soluble and highly polar interferences. I have revised and marked red in the manuscript (Lines 275-277).
References:
[1]. Yıldırım, S.; Fikarová, K.; PilaÅ™ová, V.; Nováková, L.; Solich, P.; Horstkotte, B. Lab-in-Syringe Automated Protein Precipitation and Salting-out Homogenous Liquid-Liquid Extraction Coupled Online to UHPLC-MS/MS for the Determination of Beta-Blockers in Serum. Anal. Chim. Acta 2023, 1251, 340966, doi:10.1016/j.aca.2023.340966.
[2]. Dong, H.; Guo, X.; Xian, Y.; Luo, H.; Wang, B.; Wu, Y. A Salting Out-Acetonitrile Homogeneous Extraction Coupled with Gas Chromatography–Mass Spectrometry Method for the Simultaneous Determination of Thirteen N -Nitrosamines in Skin Care Cosmetics. J. Chromatogr. A 2015, 1422, 82–88, doi:10.1016/j.chroma.2015.10.044.
[3]. Leite, M.; Freitas, A.; Barbosa, J.; Ramos, F. Comprehensive Assessment of Different Extraction Methodologies for Optimization and Validation of an Analytical Multi-Method for Determination of Emerging and Regulated Mycotoxins in Maize by UHPLC-MS/MS. Food Chem. Adv. 2023, 2, 100145, doi:10.1016/j.focha.2022.100145.
Comment 11: What is the rationale behind using 2mM ammonium acetate?
Response 11: Thanks for your kind reminder. In this experiment, we refer to the methods reported by Li and Temerdashev et al. [1,2]. The addition of ammonium acetate in the mobile phase can maintain a certain pH and ionic strength of the mobile phase, which can promote the protonation of hydroxyl groups in the reactants, which can increase the rate of the reaction, reduce the trailing, and improve the peak shape. Pre-experiments were conducted to optimize the amount of ammonium acetate added, and it was found that 2 mM addition gave the best results.
References:
[1]. Li, J.; Xu, A.; Xue, J.; Qian, W.; Xu, P.; Hu, Z.; Chen, C.; Wu, C. Development and Validation of a Deep Eutectic Solvent-Assisted Liquid-Liquid Extraction Method for Simultaneous Quantification of Six Steroid Hormones in Serum by Liquid Chromatography-Tandem Mass Spectrometry. J. Chromatogr. A 2023, 1710, 464413, doi:10.1016/j.chroma.2023.464413.
[2]. Temerdashev, A.; Azaryan, A.; Dmitrieva, E. Meldonium Determination in Milk and Meat through UHPLC-HRMS. Heliyon 2020, 6, e04771, doi:10.1016/j.heliyon.2020.e04771.
Comment 12: Provide the need for the conversion multiplier in Equation (2).
Response 12: Thanks for your kind reminder. After 5 g had been extracted by 10 mL of extract, 4 mL of extract was concentrated to 0.5 mL, and the instrument's assay concentration was four times the sample concentration. In order to facilitate the reader's calculations, the multiplier needs to be converted in Eq. (2).
Comment 13: In the conclusion section, please add a sentence about QVet-NM extraction used for the purification of samples.
Response 13: Thanks for your kind reminder. The use of QVet-NM for sample purification has been illustrated by adding "Following extraction, the extracts were purified using the QVet-NM one-step column, which greatly improved the detection efficiency and consumed less organic solvents." (Lines 349-351) to the conclusion section of the article, which has been highlighted in red in the manuscript.
Comment 14: Keep a single approved name of metenolone i.e., either methenolone or metenolone to avoid confusion for the readers and maintain consistency.
Response 14: Thanks for pointing out the mistake. We apologize for the spelling errors, which have been corrected and red-flagged in the manuscript and spelling checks have been made for the other 11 compounds.
Comment 15: The full name of the term ‘Qvet-NM’ should be mentioned. Also, mention the source of procurement of SPE columns in Section 3.1.
Response 15: Thanks for your suggestion. The full name of QVet-NM has been indicated (Lines 163-164) and the source of procurement has been stated in 3.1 (Lines 235-236).
Comment 16: In Section 2.4 and Figure 3, change the term ‘mass spectrogram’ to ‘extracted ion chromatogram’.
Response 16: Thanks for your very careful advice. The mass spectra have been changed to extracted ion chromatograms in 2.5 (Line 213-214) and Figure 5 (Line 218-221) and have been labeled in red in the manuscript.
Comment 17: How was the uniformity in sample volume maintained under a gentle blow of nitrogen?
Response 17: Thanks for your question. First of all, when nitrogen blowing, make sure that the flow rate is appropriate and uniform to prevent liquid spattering. Furthermore, the concentration volume was kept consistent by means of a nitrogen blowing tube (10 mL), and the nitrogen was blown to a volume of 50 μL, as shown in Figure 3.
Figure 3. Schematic of nitrogen blow concentration volume.
Comment 18: Slight dryness is a misleading terminology, mention the time frame for which samples were kept for evaporation to concentrate.
Response 18: Thanks for your suggestion. We apologize for the inappropriate representation of the article. The filtrate was nitrogen blown to 50 μL for 25-30 min at 45 °C in a water bath. The statement has been changed in Section 3.5 (Lines 282-284) and highlighted in red in the manuscript.
Comment 19: Please provide the precursor and product ion spectra (qualifier and quantifier) of all the androgenic steroids in a supplementary information.
Response 19: Thank you for your kind reminder. We have submitted the precursor and product ion spectra (qualitative and quantitative) for all androgenic steroids in the supplement. The file name is Precursor and product ion spectra (qualitative and quantitative) of all androgenic steroids.

Reviewer 2 Report
Comments and Suggestions for Authors
Reviewer report on manuscript Molecules-2745114
The submitted experimental work aimed towards the development and validation of a UHPLC-MS/MS method for the determination of anabolic steroids in livestock and poultry meat using one-step solid phase extraction.
I read the manuscript carefully. In general terms, the manuscript is well-written and the method is well-validated. However, what I missed in the submitted article is the novelty of the proposed approach compared to the existing UHPLC-MS/MS methodologies. What are the advantages of the proposed methods compared to the existing knowledge? This point must be clearly justified in the introduction. The potential advantages of the proposed SPE technique over the other reported methods should be stated.
Other comments
1) The recovery and %RSD values must be given with one decimal point. Make this change in the entire manuscript.
Comments on the Quality of English LanguageSatisfactory
Author Response
Comment 1: What are the advantages of the proposed methods compared to the existing knowledge? This point must be clearly justified in the introduction. The potential advantages of the proposed SPE technique over the other reported methods should be stated.
Response 1: Thanks for your suggestions. We have added the explanation about the novelty and advantages of the proposed methods compared to the current methods in the revised introduction part (Lines 69-75). In addition, the potential advantages of the SPE technique compared to other reporting methods, and the modifications have been highlighted in red in the revised manuscript (Lines 55-58).
Comment 2: The recovery and %RSD values must be given with one decimal point. Make this change in the entire manuscript.
Response 2: Thanks for your kind reminder. It has been revised and marked red in the manuscript.

Round 2
Reviewer 1 Report
Comments and Suggestions for Authors
The authors have now answered all the concerns raised in the first revision.
Reviewer 2 Report
Comments and Suggestions for Authors
The authors addressed properly my previous comments. Therefore, I recommend its acceptance for publication.